# The Cow’s Milk Related Symptom Score: The 2022 Update

**DOI:** 10.3390/nu14132682

**Published:** 2022-06-28

**Authors:** Yvan Vandenplas, Katerina Bajerova, Christophe Dupont, Philippe Eigenmann, Mikael Kuitunen, Rosan Meyer, Carmen Ribes-Koninckx, Silvia Salvatore, Raanan Shamir, Hania Szajewska

**Affiliations:** 1Vrije Universiteit Brussel (VUB), UZ Brussel, KidZ Health Castle, Laarbeeklaan 101, 1090 Brussels, Belgium; 2Department of Pediatrics, University Hospital Brno and Masaryk’s University, 625 00 Brno, Czech Republic; bajerova.katerina@fnbrno.cz; 3Department of Internal Medicine, Geriatrics and Practical Medicine, University Hospital Brno and Masaryk´s University, 625 00 Brno, Czech Republic; 4Ramsay Group, France et Clinique Marcel Sembat, Paris Descartes University, Boulogne-Billancourt, 75004 Paris, France; christophe.dupont@wanadoo.fr; 5Pediatric Allergy Unit, University Hospitals of Geneva, 1205 Geneva, Switzerland; philippe.eigenmann@hcuge.ch; 6Children’s Hospital, University of Helsinki and Helsinki University Hospital, 00290 Helsinki, Finland; mikael.kuitunen@hus.fi; 7Department Paediatrics, Imperial College London, London SW7 2BX, UK; info@rosan-paediatricdietitian.com; 8Department Dietetics, Winchester University, Winchester SO23 4NR, UK; 9Department Medicine, KU Leuven, 3001 Leuven, Belgium; 10Pediatric Gastroenterology, La Fe University Hospital, Instituto de Iinvestigacion Sanitaria La FE Valencia, 46026 Valencia, Spain; ribes_car@gva.es; 11Department of Paediatrics, University of Insubria, 21100 Varese, Italy; silvia.salvatore@uninsubria.it; 12Institute of Gastroenterology, Nutrition and Liver Diseases, Schneider Children’s Medical Center, Lea and Arieh Pickel for Pediatric Research, Sackler Faculty of Medicine, Tel-Aviv University, Tel-Aviv 69978, Israel; raanan@shamirmd.com; 13Department of Paediatrics, The Medical University of Warsaw, 02-014 Warsaw, Poland; hszajewska@wum.edu.pl

**Keywords:** cow’s milk allergy, cow’s milk-related symptom score, CoMiSS, functional gastrointestinal disorder, infant feeding

## Abstract

CoMiSS^®^ was developed 7 years ago to increase the awareness of health care professionals towards the possibility that symptoms presented by infants could be related to cow’s milk. While CoMiSS was conceived mostly on theoretical concepts, data is now available from 25 clinical trials. Based on this extensive research using the tool since 2015, we aim to propose an updated CoMiSS. The evidence was reviewed, debated and discussed by 10 experts, of whom seven were part of the original group. The panel concluded that the cut-off previously proposed to indicate the likelihood that symptoms may be cow’s milk related should be lowered from ≥12 to ≥10. Data in healthy infants > 6 months are missing. Since the Brussels Infant and Toddlers Stool Scale (BITSS) was recently developed for non-toilet trained children, the Bristol Stool Scale was changed to the BITSS without changing the impact of stool characteristics on CoMiSS. Overall, CoMiSS raises awareness that symptoms might be cow’s milk related. New studies are needed to determine if the change in cut-off and other small adaptions improve its sensitivity and specificity. Data for CoMiSS is still needed in presumed healthy infants between 6 and 12 months old. There may also be regional differences in CoMiSS, in healthy infants as well as in those with cow’s milk allergy. Finally, we emphasize that CoMiSS is an awareness tool and not a diagnostic test.

## 1. Introduction

The diagnosis of cow’s milk allergy (CMA), one of the most common food allergies in early childhood, remains a challenge in clinical practice [1]. While the immediate symptoms of immunoglobulin E (IgE) mediated CMA are easily recognized, the diagnosis of non-IgE mediated CMA is often a challenge because of the delayed symptom onset and overlap with common functional gastrointestinal disorders manifestations such as infantile colic, gastro-esophageal reflux (disease) or infections [2,3].

To assist in identifying cow’s milk (CM) related symptoms, a “Symptom-Based Score” (SBS) was developed to include infants with comparable severity of symptoms suspected to be CM-related as part of a multicenter trial (14 hospital sites) comparing the efficacy of two extensively hydrolyzed formulas [4]. The SBS consisted of a rapid and easy-to-use questionnaire assessing stool pattern, presence and intensity of crying/irritability and regurgitation, as well as skin and respiratory manifestations (Table 1). The total score ranges from 0 to a maximum of 33, with an arbitrary proposed cut-off value of ≥12, which was used as inclusion criterion in the original trial for which the tool was developed [4]. The cut-off was decided by unanimous consensus between all co-investigators [4].

Following on from this study, the SBS was discussed by an international group of experts with clinical experience in managing infants with CMA and was converted without any change into the Cow’s Milk-related Symptom Score (CoMiSS^®^) [5].

The intention of CoMiSS is to increase the awareness of health care professionals (HCPs) towards the possibility that symptoms presented by the infant could be related to CM intake. Experience and published data have suggested that the score may be of particular interest to suspected non-Immunoglobulin E (IgE) mediated allergy but does also include some symptoms of IgE mediated allergy [1]. In fact, CoMiSS does contain symptoms which are IgE mediated, such as urticaria but also vomiting, diarrhea and other gastrointestinal (GI) symptoms that can be either IgE as non-IgE mediated (Table 2: signs and symptoms associated with CM intake).

The goal of this manuscript is to propose an updated CoMiSS based on the acquired information obtained by the extensive research using the tool since its first publication in 2015. In the last seven years, 11 studies documented that a score of ≥12 is predictive of a favorable response to a CM-free diet, showing an estimated sensitivity between 20% and 77%, a specificity of 54% to 92% for the diagnosis of CMA [1]. The sensitivity of 20% for the cut-off ≥12 was reported in a study including mainly infants with hematochezia; the highest sensitivity was found in a study with infants presenting with symptoms suggestive of CMA [1]. The awareness of symptoms of CMA might play a role in patient selection and influence sensitivity and specificity [1]. The majority of the infants were formula fed, although some were mixed or exclusively breastfed [1]. The documented sensitivity of the cut off score (≥12) for CMA was deemed to be insufficient because CoMiSS was originally intended as an awareness tool. Following the accumulated experience with CoMiSS, a lower cut-off has been proposed. This is because the 95th centile of CoMiSS in presumed healthy infants was estimated at 9 [6], and among 13 presumed healthy infants with score ≥10 in one study, 10 cases (76%) were actually diagnosed with CMA [7]. Moreover, three studies assessing sensitivity and specificity to CMA of CoMiSS among symptomatic infants using the Receiver Operating Curve (ROC) have proposed an even lower cut-off, one with ≥9 [8] and the other ≥6 [9].

## 2. Methods

Seven years after its first publication, a meeting was organized attended by 10 clinicians with expertise in managing children with gastrointestinal problems and/or atopic diseases to review and discuss published data on CoMiSS. Seven of them were part of the initial group. The goal of the meeting was to discuss if the opportunity had arisen to update CoMiSS.

The authors discussed the difficulties to diagnose CMA and the contribution of CoMiSS so far. Symptoms included in CoMiSS were discussed. It was debated if some symptoms should be taken out, or if others should be added. No universal and consistent guidelines exist on the methodology of conducting e-Delphi studies which result in large variability in its execution. Statements were formulated. The draft document containing the list of statements formulated by the core group was circulated by email to all group members. Each member was asked to vote by marking “agree”, “abstain” or “disagree” beside each statement. Each member was given the opportunity to provide comments and suggest different wording. Anonymity was not retained. Eighty percent agreement from the group was required in order to accept or omit a statement during development of the final document. If a statement would not obtain 80% agreement, it was planned to modify the statement according to feedback provided by the group members and sent to the group for a second round.

## 3. A Score as a Diagnostic or Awareness Tool?

IgE mediated CMA is relatively easy to recognize and diagnose because of the short interval time between CM ingestion and appearance of symptoms. Moreover, specific IgE (sIgE) levels and skin prick tests (SPT) are supportive diagnostic tests [3]. Children with IgE mediated CMA are known to be at increased risk to develop later in life other conditions such as asthma and rhinitis, the so-called “atopic march” [10]. Symptoms, clearly related to milk ingestion, may occur without any demonstrable immune mediated allergic mechanism or may be due to non IgE mediated CMA. For the sake of simplicity, we will thereafter mention all of them as CMA [11,12]. In most of the articles included in the review paper by Meyer et al. CMA was confirmed by a hospital challenge or reintroduction of CM at home [11]. Except for a clear anaphylactic reaction, none of the symptoms associated with CMA (Table 2) are specific and can be caused by non-immunological mechanisms, such as infection or functional gastro-intestinal disorders (FGIDs). Therefore, no scoring system or tool will ever be able to reliably pick-up all patients suffering from CMA.

Children with non-IgE mediated CMA or non-allergic milk-related symptoms present with GI symptoms (gastroesophageal reflux, vomiting, diarrhea, constipation, flatulence, blood and mucus in stools, colic, general symptoms, faltering growth) more frequently than those who do not suffer from CMA (65% vs. 42%; <0.001) [13]. In 28% of the infants with CMA, symptoms are present in more than one organ system [13]. According to the same author, skin and respiratory tract manifestations occur frequently in infants with CMA. Sorensen at al. classified irritability as part of GI symptoms, occurring in 24% of the infants with CMA (vs 9.5% in non-CMA infants). If irritability would be classified as “other” organ system, then 52% of CMA-infants would present symptoms in other organ systems [13], bringing the prevalence close to the 59% reported by Sladkevicus et al. [14]. Most patients with CMA present with a combination of symptoms, often spread over different organ systems.

Most infants with FGIDs also present with a combination of different FGIDs [15,16,17,18]. However, in FGIDs the symptoms are limited to the GI tract and to general manifestations (irritability, excessive crying). Thus, when GI and/or general manifestations are combined with skin and/or respiratory tract manifestations, the presence of CMA more likely. Nevertheless, the clinical finding may also occur by coincidence given the chance of non immune related skin manifestations and the high prevalence of symptoms in the respiratory tract in childhood. Nevertheless, almost 70% of children with CMA report upper respiratory tract infections [19,20]. Sorensen et al. found a higher prevalence of otitis media (25 vs. 19%; <0.01; rate 0.51 vs. 0.36; <0.01) and respiratory tract infections (89% vs. 82%; rate 6.88 vs. 5.03 (<0.01) and asthma (7.1 vs. 3.8%;<0.01) [13]. Nonetheless, children with non-IgE mediated GI allergies, experience atopic dermatitis in around 40–50% of cases [12,19,21], though the role of CM in atopic flares may largely be non-allergy related. Therefore, any score may inappropriately label some infants with multiple symptoms as “allergic” and may miss “allergy” in some infants with less or different symptoms.

Due to the impact on long-term health, CMA should only be suspected on the basis of a complete history, physical examination, and anthropometric assessment. The management of CMA as well as the indication, performance and interpretation of a food challenge is not described herein as this is beyond the scope of this manuscript. However, it is important to note that food allergic infants who are not suitably nutritionally managed and/or follow inappropriate elimination diets, may show faltering growth in the first years of life and reduced height in adulthood [22]. Associations between CMA and other clinical entities may not be obvious and thus an allergy focused history and examination to assess for the possibility of CMA is essential.

Ultimately, it is all about “probability”: CoMiSS must be considered as an awareness tool indicating likelihood but is not intended for a diagnostic use. For an awareness tool, sensitivity is more important than specificity as it should identify most if not all cases. Also setting the cut-off too low will cause unnecessary diagnostic work-up of cases with low probability of CMA.

### 3.1. The Updated CoMiSS

All authors voted on the different statements listed in Table 3. The modified Delphi process to establish consensus on the statements was used. Table 4 represents the updated CoMiSS.

### 3.2. Crying/Irritability

Parental perception of severity and duration of crying is subjective. Measuring “irritability” or “crying” with objective tools and diary recording would result in more exact data. However, technical limitations render such options unrealistic, as there is no device on the market that accurately measures infant crying. Depending on how much a baby cries, the parents’ quality of life can be affected, which may cause over-reporting [23]. Therefore, on this basis it was decided not to change the scoring of crying time. It was also re-stressed that the crying should exist for at least one week to avoid scoring the acute causes of intense bouts of crying such as those caused by infections and trauma or surgical conditions (i.e., hernia, invagination, etc.). 

### 3.3. Regurgitation

The scoring for the severity of regurgitation was adopted from an arbitrary scale developed for the first clinical trial evaluating the efficacy of a commercialized thickened anti-regurgitation formula [24] and, since then, this score was used in many trials. As for mechanistic factors, regurgitation is distinguished from vomiting by the absence of a central nervous system (CNS) emetic reflex, retrograde upper intestinal contractions, nausea, and retching [25]. Vomiting is defined as an expulsion with force of the refluxed gastric contents from the mouth [25] and is a coordinated autonomic and voluntary motor response, causing forceful expulsion of gastric contents [25]. In food-protein induced enterocolitis syndrome (FPIES), there is chronic vomiting in the chronic presentation and forceful vomiting in acute FPIES [26]. In both cases, vomiting is related to a CNS emetic reflex triggered by the presence of milk in the digestive tract. Since the difference between vomiting and regurgitation is not easy for parents and HCPs to distinguish, the score for regurgitation includes vomiting. The CoMiSS-group voted in consensus that there was no need to update the regurgitation-severity score. It was decided to clarify that regurgitation should exist for at least one week to exclude acute cause of regurgitation such as infections.

### 3.4. Stools

The Brussels Infant and Toddlers Stool Scale (BITSS) has been recently reported to represent stool consistency in a more appropriate way in non-toilet-trained children than the Bristol Stool Scale [9]. The pros and cons of the inclusion of other symptoms such as hematochezia has been debated as well.

The original SBS and consequently CoMiSS did use the Bristol stool scale (BSS) to describe stool consistency [27]. However, there is consensus that the Bristol stool scale, originally developed to describe transit time in adults, is not appropriate to describe stool consistency in non-toilet trained children [28]. Although the BSS has not been validated in children, the modified Bristol Stool Form Scale for Children (mBSFS-C) has been validated [29,30]. The mBSFS-C has the same text and pictorial descriptors as the BSS, but does not include types 3 and 5 from the original scale [29]. The mBSFS-C has already shown good concurrent validity and inter-rater reliability in estimating stool form amongst children and pediatric gastroenterologists [29,31].

The Brussels Infant and Toddler Stool Scale (BITSS) was developed to describe stool consistency in the diapers of non-toilet trained children [9]. A study has been performed to indicate that BITSS was appropriate, and a digital tool was developed avoiding subjective interpretation by parents [9]. The following weighting was given to the different stool consistencies described in BITSS: a score of 4 for hard, 0 for formed, 4 for loose and 6 for watery stools. A pair-wise comparison was made between BITSS and the score given to the stool consistency according to the BSS in the original CoMiSS [32]. The analysis of the pairwise scores showed that CoMiSS with the BITSS did not change the score compared to the original CoMiSS with the BSS in 565/844 (67%) subjects. CoMiSS with the BITSS compared to original CoMiSS with the BSS changed only in 2/844 (0.24%) cases from at-low-risk to at-high-risk when the cut-off ≥12 was used, and in 3/844 (0.36%) when cut-off ≥10 was applied. CoMiSS with BITSS changed only in 1/844 (0.12%) infants from at-high-risk to at-low-risk for cut-off ≥12 and in 0/844 (0%) cases for cut-off ≥10. Overall, the difference between the two total scorings was statistically significant (*p* = 0.001) but not in subjects (N = 304) with CoMiSS-BSS ≥6 (*p* = 0.81). Fifty-seven of these infants had a CoMiSS of ≥10 and 23 a CoMiSS of ≥12. Changing BSS to BITSS had no impact on CoMiSS for those infants with a score ≥6 [32]. As a consequence, it was considered to replace BSS by BITSS in the updated CoMiSS. BSS and BITSS can be used interchangeable since there is no impact of CoMiSS. Also, stool consistency should be stable for at least one week.

### 3.5. Hematochezia

Food protein-induced allergic proctocolitis (FPIAP; formerly known as allergic or eosinophilic proctocolitis) often presents as rectal bleeding, hematochezia or persistent mucus-streaked diarrhea in an otherwise healthy young infant [33]. FPIAP prevalence estimates range widely from 0.16% in healthy children to 64% in patients with blood in stools [34,35]. This disease usually manifests within the first weeks of life and resolves by late infancy in most cases. It is characterized by inflammation of the distal colon in response to one or more food proteins through a mechanism that does not involve IgE. The presence of anal fissures in young infants raises the possibility of being related to CMA [36].

FPIAP is a benign, easily recognized syndrome that should not be further investigated and may not need treatment [37]. In apparently healthy infants, hematochezia is more frequent in breastfed than in formula fed infants [37]. However, hematochezia can also be the presenting symptom of more severe disease such as late onset necrotizing enterocolitis. Infectious gastro-enteritis is a much more frequent cause of hematochezia than CMA.

In apparently healthy infants, a “wait and see” approach has been suggested for one month by the European Academy of Allergy and Clinical Immunology guidelines for breastfed infants with suspected non-IgE mediated food allergies [37]. The presence of important quantities of blood in the stools in a sick infant (painful abdomen, vomiting, pallor, lethargy, hypo- or hyperthermia, …) requires immediate referral and action, what may in some cases be withdrawal of CM from the diet. Therefore, our group decided to not include hematochezia in CoMiSS, but instead indicate that if it occurs in sick infants an immediate and urgent referral and full diagnostic work up are required.

### 3.6. Food Protein-Induced Enterocolitis

Food protein-induced enterocolitis syndrome is a non–IgE-mediated food allergy phenomenon with CM being one of the most commonly reported triggers [38,39]. 

FPIES is still underdiagnosed despite it being considered a potential medical emergency. FPIES typically presents in infancy with repetitive protracted emesis approximately 1 to 4 h after food ingestion [38,39]. However, since the presenting symptoms of FPIES are part of CoMiSS, this tool should pick-up FPIES. 

### 3.7. Dermatological Symptoms 

CoMiSS includes an easy-to-use scoring for the assessment of atopic dermatitis [40]. Although worsening of existing eczema may indicate the involvement of CM, we did not include this aspect in CoMiSS since the aspect of worsening eczema would be based on subjective parental reporting. Angioedema and urticaria are two typically IgE mediated symptoms. Angioedema was not included in the original CoMiSS, whilst urticaria was part of it. Since many HCPs may not consider urticaria as a symptom associated with CM intake, the group decided to keep urticaria in CoMiSS, and not to change the weight of scoring accorded. It was also decided to include angioedema in the updated CoMiSS, in combination with urticaria as angioedema and urticaria frequently co-exist and result from similar mechanisms. If present, urticaria and angioedema both score 6; the combination of both also scores 6. The same score for this part of the score was kept as in the original. However, if urticaria/angioedema can be directly related to cow’s milk (e.g., drinking milk and no intake of another food item), this is strongly suggestive of a CMA, and may not need a further cow’s milk challenge. Furthermore, whilst eczema should exist for at least one week, urticaria and angioedema are acute symptoms.

### 3.8. Respiratory Symptoms

The inclusion of respiratory symptoms in the CoMiSS has been debated since the vast majority of respiratory symptoms in infants are caused by (viral) infections. Nevertheless, respiratory symptoms are also listed by major guidelines as being possibly related to CMA [41]. The link of respiratory symptoms to CMA results from acute IgE-mediated symptoms observed during food challenges, and for these, there is no doubt that they are linked to food allergy. Although rarely occurring in isolation, respiratory symptoms are of particular importance to patients with CMA as they are associated with severe clinical manifestations [41]. During food challenges (to CM) rhinitis occurs in 70% of cases and asthma in up to 8% [41]. CoMiSS mentions that the symptoms should be “chronic, and not related to infection”. Distinguishing between respiratory symptoms caused by infection or allergy remains challenging during infancy. Although most frequently respiratory symptoms during a challenge are of the acute type, it was estimated that acute respiratory symptoms during infancy are by far more frequently caused by infection than by an allergic reaction. Therefore, the respiratory symptoms were included in the original CoMiSS but given less weight than the other parameters. Consensus was achieved to keep the scoring for respiratory symptoms that are present for at least a week, and to not change the weight given to it.

### 3.9. Family History

History of atopic disease in first-degree family members, diagnosed by a health care professional, has long time been recognized as a risk factor for atopic disease in the offspring [42]. Having a sibling with allergic disease was reported to almost double the risk of food allergy in the child compared with having no family history of allergy, even in the absence of parental history of allergy (9.6% vs. 5.6% in children with siblings, *p* = 0.025) [43]. However, we decided to not include family history in the updated CoMiSS because infants with no family history can also develop allergies [44]. Moreover, reliable reporting of family history for allergy would require education of parents and a confirmed diagnosis. Noteworthy, both the Australian and the UK guidelines on allergy prevention no longer consider family history as a risk factor [43,44].

### 3.10. Cut-Off and Age

A large study to determine CoMiSS in 891 apparently healthy infants younger than 6 months revealed an overall median and mean (SD) CoMiSS of respectively, 3.0 and 3.7 (2.9). The 95th percentile was 9 [6]. Therefore, a cut-off score of ≥9 or ≥10 has been tested. Experience from these studies reported a sensitivity of 84 to 88% and a specificity of 85% [1]. The group decided to propose a cut-off of ≥10, and to recommend a preferred age limit of 6 months since there are currently no data in healthy infants beyond this age. However, there was consensus within the group that CoMiSS can be used up to the age of one year, but not beyond.

### 3.11. Anaphylaxis

Anaphylaxis represents the severe end of the spectrum of allergic reactions. Anaphylaxis is a serious systemic hypersensitivity reaction that is usually rapid in onset and may cause death [45]. Severe anaphylaxis is characterized by potentially life-threatening compromise in breathing and/or the circulation and may occur without typical skin features or circulatory shock being present. If anaphylaxis had to be included in CoMiSS, it should get a score above the cut-off. Since anaphylaxis is an alarm symptom requiring immediate action and referral, and since CoMiSS is an awareness tool (and not a diagnostic tool), there was consensus not to include this in the CoMiSS. The group therefore decided that the updated tool should clearly indicate that it is not intended for infants with severe and life-threatening symptoms clearly indicating CMA and therefore requiring immediate referral. 

### 3.12. Failure to Thrive

Like anaphylaxis, failure to thrive (FTT) is an alarm symptom requiring referral and a broad diagnostic work-up for full understanding of the cause. Since CoMiSS is an awareness tool, and multiple factors and underlying disease may determine FTT, this was not included in CoMiSS.

## 4. Results

Statements were formulated and voted on with “agree” or “disagree”, offering the opportunity to make comments.

All statements were approved within the first voting round since 80% agreement was reached on all statements. Whether CoMiSS should be preferable used ≤6 months or ≤12 months was debated amongst the group. The argument for ≤6 months is that scores in an apparently healthy population is limited to 6 months of age. The argument for ≤12 months is the fact that cow’s milk is introduced only after the age of 6 months in many infants. CoMiSS should not be used beyond the age of 12 months. This highlights that the priority for future research is the determination of CoMiSS in an apparently healthy population of 6 to 12 month old infants. It cannot be excluded that the cut-off in infants ≤6 months and 6–12 month old infants might be different. Therefore, the group came to consensus that CoMiSS should preferably not be used in children older than 1 year, and this was indicated on the updated tool.

Since all statements were approved, CoMiSS was updated accordingly (Table 4).

## 5. Discussion

Reviewing the abstract first published from 2015, CoMiSS did well in achieving its objectives. The abstract contained the following phrase “The Cow’s Milk-related Symptom Score (CoMiSS), which considers general manifestations, dermatological, gastrointestinal and respiratory symptoms, was developed as an awareness tool for cow’s milk-related symptoms” [4]. Evidence has shown that CoMiSS is indeed a reliable awareness tool [1]). The abstract continued: “CoMiSS can also be used to evaluate and quantify the evolution of symptoms during therapeutic interventions” [4]. Again, evidence confirmed this proposal [1]. And finally, the 2015-abstract concluded “CoMiSS does not diagnose CMA and does not replace a food challenge.” This is still valid today. 

CoMiSS was developed as a practical clinical tool with the goal to increase the awareness of HCP for the presence and intensity of clinical manifestations possibly related to CM consumption [5]. During the past seven years, 25 original studies using CoMiSS were published [1]. Infants exhibiting symptoms possibly related to CM, present with a higher median CoMiSS (median 6 to 13; mean 11.2 to16.2; 16 studies) than apparently healthy infants (median 3 to 4; and mean 3.6 to 4.7; 5 studies) [1]. In children with CMA, 11 studies found that a CoMiSS of ≥12 predicted a favorable response to a CM-free diet; however, sensitivity (20% to 77%) and specificity (54% to 92%) varied substantially, related to large differences in inclusion criteria and primary endpoints [1]. A low CoMiSS (< 6) was predictive for the absence of CMA [1]. 

The score obtained by HCPs and by caregivers is comparable, suggesting that no special training to use the tool is required [46]. Intra-rater reliability was high with very low variability (ICC 0.93; 95% CI 0.90–0.96; *p* < 0.001) in repeated assessments [46]. 

Delayed diagnosis of CMA is well known to be associated with nutritional deficiencies, bone density and faltering growth [22,47,48,49,50]). The risk for over- as well as under-diagnosis of CMA should be minimized because both are associated with a negative impact on health and QoL [1]. Elimination diets, especially when long lasting, can lead to nutritional inadequacies [51,52] and by reducing the variety of the diet they can have long-term effects on eating behavior and taste preferences. Moreover, the economic impact of diet on health care system and/or family should also be considered. A sensitive and specific awareness tool seems to be the best way how to approach a possible diagnosis CMA. Since CoMiSS is developed as an awareness tool, sensitivity has priority above specificity although of course it would be ideal to see both above 90%.

The panel members confirmed in consensus that experience has shown that CoMiSS, as predicted, cannot be used as a stand-alone diagnostic tool. However, CoMiSS is a useful and handy tool intended to increase awareness of HCPs about the symptomatology of CMA. Given the specific nature and variability of symptoms, diagnosis should always be the result of clinical interpretation of the outcome of a 2–4 week diagnostic elimination diet followed by a challenge test [53]. However, the exception would be in the case of clear immediate IgE type of reactions such as is anaphylaxis, urticaria and angioedema. 

Based on the data published, the panel agreed to decrease the cut-off from ≥12 to ≥10. Thus: if CoMiSS is ≥10, we recommend starting a diagnostic elimination diet for 2–4 weeks, followed by a challenge test to confirm or refute the diagnosis Moreover, a CoMiSS < 10 does not exclude the diagnosis of CMA. Conversely, a score of ≥10 is only indicative of an increased likelihood that CMA might exist, with no confirmed diagnosis. Although a positive SPT or elevated sIgE levels increase the likelihood of CMA, these test are only indicative of sensitization and require an OFC to confirm the diagnosis [10,53]. Most participants estimate that CoMiSS would be a useful tool up to the age of 12 months. However, the presumed healthy population only included infants between 1 and 6 months old. As long as CoMiSS in a healthy 6–12 months population is not known, it was finally concluded to recommend a preferred use up to 6 months, with an extended use up to 12 months. As a consequence, determination of CoMiSS in a presumed healthy population of 6–12 month old infants is the most urgently needed future research topic. Indeed, CM is in many infants introduced after the age of 6 months. According to the origin country or region of the published studies, there may also be regional differences in CoMiSS in healthy infants and infants suspected to have CMA. This hypothesis needs further evaluation as well.

One panel member proposed the removal of the respiratory symptoms because they are frequently of infectious origin. Another member also was in favour of this but finally agreed to keep respiratory symptoms because removal would mean a substantial change of the tool thus implying that the existing information available on CoMiSS would be of no further use. One panel member voted to remove the dermatological symptoms with the argumentation that focus of CoMiSS should only be on non-IgE mediated GI symptoms, but a large majority decided to keep skin symptoms in the tool. Urticaria was proposed to be removed by one member because it is a diagnosis of CMA, not an awareness if it is clearly temporally related to the intake of milk. One other member proposed to remove it because urticarial symptoms are frequently caused by viral infections or other foods.

Another change that was introduced, was the replacement of the BSS by the BITSS. The images of the BSS are not representative for the stools of non-toilet trained children. The BITSS was specifically developed for this age group [9]. Moreover, an artificial intelligence tool was developed using BITSS, that will decrease the subjectivity of the interpretation of the stool consistency [54]. One panel member proposed to not introduce BITSS because it does not change CoMiSS with BSS. Indeed, the scoring according to BITSS was evaluated with no impact on CoMiSS values compared to the original CoMiSS with BSS. This was the intention, because any change with an impact on CoMiSS would mean that the accumulated experience and published data can no longer be used. 

## 6. Conclusions

In conclusion: CoMiSS is a useful awareness tool for evaluating cow’s milk-related symptoms in otherwise healthy infants by preference less than 6 months old, although there was consensus to allow extension up to 12 months. A cut-off of ≥10 may be suggestive of CMA, and CMA is unlikely if CoMiSS is ≤6. BSS and BITSS can be used interchangeably. In the updated tool, angioedema was added to the symptoms and further clarification was provided regarding the need for urgent action.

## Figures and Tables

**Table 1 nutrients-14-02682-t001:** Symptom-based-score and CoMiSS (adapted from Refs [2,3]).

Symptom	Score	
Crying (°)	0	≤1 h/day
	1	1–1.5 h/day
	2	1.5–2 h/day
	3	2 to 3 h/day
	4	3 to 4 h/day
	5	4 to 5 h/day
	6	≥5 h/day
Regurgitation	0	0–2 episodes/day
	1	≥3–≤5 of small volume
	2	>5 episodes of >1 coffee spoon
	3	>5 episodes of ± half of the feed in <half of the feeds
	4	continuous regurgitations of small volumes >30 min after each feed
	5	regurgitation of half to complete volume of a feed in at least half of the feeds
	6	regurgitation of the complete feed after each feeding
Stools (Bristol scale)	4	type 1 and 2 (hard stools)
	0	type 3 and 4 (normal stools)
	2	type 5 (soft stool)
	4	type 6 (liquid stool, if unrelated to infection)
	6	type 7 (watery stools)
Skin symptoms	0 to 6	Atopic eczema
		Head neck trunk Arms hands legs feet
		Absent 0 0
		Mild 1 1
		Moderate 2 2
		Severe 3 3
	0 or 6	Urticaria (no 0/yes 6)
Respiratory symptoms	0	no respiratory symptoms
	1	slight symptoms
	2	mild symptoms
	3	severe symptoms

Legend: (°) Crying was only considered if the child was crying for one week or more, assessed by the parents, without any other obvious cause.

**Table 2 nutrients-14-02682-t002:** Signs and symptoms eventually associated with cow’s milk intake *.

General	Excessive crying, irritability *
	Failure to thrive
	Iron deficiency anemia
Gastro-intestinal °	Dysphagia
	Regurgitation, vomiting °, GER
	Diarrhea
	Constipation ± perianal rash
	Anal fissures
	Blood loss/Hemotochezia °
Respiratory °	Rhinitis sneezing
	Cough
	Eye swelling and redness
	Wheezing
Skin	Erythema, redness
	Eczema (atopic dermatitis)
	Worsening of existing eczema
	Urticaria °
	Angioedema

Legend: *: none of the symptoms are specific; crying and irritability are classified under general manifestations since as it’s origin can as well be gastro-intestinal as dermatological caused by itching; °: chronic and unrelated to infection; GER: gastro-esophageal reflux.

**Table 3 nutrients-14-02682-t003:** Statements and voting results regarding the updated CoMiSS.

	Symptom	Agree	Disagree/Abstain	Comment
1	Many signs and symptoms of CMA can be seen in both IgE as well as non-IgE mediated disease	10		
2	Anaphylaxis should not be part of CoMiSS	10		
3	Failure to thrive should not be part of CoMiSS	10		
4	Hematochezia should not be part of CoMiSS	10		
5	CoMiSS should preferably be used in infants ≤6 months	9	1/0	<1 year of age
6	A cut-off of ≥10 is suggested as the new cut off value for the risk of CM-related symptoms	10		
7	“Existing since at least 1 week” should be added to all symptoms, except for urticaria and angio-edema	9	1/0	Acute urticaria is one of the most frequent signs in IgE-CMA. When recurrent or lasting more than a few hours, urticaria is most commonly not related to CMA
8	The scoring (1 to 6 in function of duration) of crying/irritability remains unchanged	10		
9	The scoring (1 to 6 in function of volume and frequency) of regurgitation remains unchanged	10		
10	The scoring (1 to 3 in function of severity) of respiratory symptoms remains unchanged	8	1/1	Remove resp symptoms. This would change CoMiSS Too many disturbing factors
11.	The scoring (1 to 6 in function of the extension and severity) of atopic eczema remains unchanged	9	1/0	Remove. We should focus only on GI symptoms
12.	Urticaria is maintained but angio-edema is added to urticaria and the same weighting in kept for both (“urticaria and/or angio-edema (No:0/yes:6)”)	8	1/1	Remove, as we should focus on GI symptoms.
13	If urticaria/angioedema can be directly related to cow’s milk (e.g., drinking milk without any other food), this is strongly suggestive of CMA, and may not need a further cow’s milk challenge	8	2/0	Delete because regards diagnosis, not awareness. Should be challenge proven. Urticaria is frequent caused by viral infection, other food.
14	The Bristol Stool Scale (BSS), developed to evaluate GI transit in adults, was replaced by the BITSS, developed to evaluate stool consistency in non-toilet trained children.	9	0/1	
15	If a weighting of 4 is given for hard, 0 for formed, 4 for loose and 6 for watery stools as described in BITSS, the impact on CoMiSS in comparison to the original scoring according to BSS remains unchanged (ref)	8	1/1	Clinical impact of CoMiSS with BITSS is not different if compared to CoMiSS with the original scoring with BSS
16	BSS (Bristol Stool Scale) and BITSS (Brussels Infant Stool Scale) can be used interchangeably	9	0/1	
17	The updated CoMiSS should continue to be used as awareness tool for evaluating cow’s milk related symptoms in otherwise healthy infants ≤6 months with a cut-off ≥10	10		
18	A CoMiSS score of ≥10 may be suggestive of CM-related symptoms	10		

Legend: BITSS: Brussels infant and toddler stool scale; BSS: Bristol stool scale; CMA: cow’s milk allergy; GI: gastro-intestinal.

**Table 4 nutrients-14-02682-t004:** Updated CoMiSS.

Symptom	Score	
Crying * assessed by parents & without any obvious cause ≥1 week	0	≤1 h/day
	1	1–1.5 h/day
	2	1.5–2 h/day
	3	2 to 3 h/day
	4	3 to 4 h/day
	5	4 to 5 h/day
	6	≥5 h/day
Regurgitation * ≥ 1 week	0	0–2 episodes/day
	1	≥3–≤5 x of volume < 5 mL
	2	>5 episodes of >5 mL
	3	>5 episodes of ±half of the feed in < half of the feeds
	4	continuous regurgitations of small volumes >30 min after each feed
	5	regurgitation of half to complete volume of a feed in at least half of the feeds
	6	regurgitation of the complete feed after each feeding
Stools * Brussels Infant and Toddlers Stool Scale (BITSS) No change ≥ 1 week	4	hard stools
	0	formed stools
	4	loose stools
	6	watery stools
Skin symptoms	0 to 6	Atopic eczema ≥1 week
		Head neck trunk Arms hands legs feet
		Absent 0 0
		Mild 1 1
		Moderate 2 2
		Severe 3 3
	0 or 6	Acute urticaria * and/or angioedema * (no 0/yes 6)
Respiratory symptoms * ≥1 week	0	no respiratory symptoms
	1	slight symptoms
	2	mild symptoms
	3	severe symptoms
**Additional information to consider**		
Worsening of existing eczema might be indicative of CMA		
If urticaria/angioedema can be directly related to cow’s milk (e.g., drinking milk in the absence of other food) this is strongly suggestive of CMA.		

Legend * in the absence of infectious disease.

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
