# Peer review of "The Cow’s Milk Related Symptom Score: The 2022 Update"

_nutrients, 2022, doi:10.3390/nu14132682_

Round 1

Reviewer 1 Report

The manuscript by Vandenplas et al. describes the updates made to CoMiSS by 10 panelists after evaluating the evidence accumulated for the use of CoMiSS as an awareness tool over the last seven years. The authors discussed individual scoring components of CoMiSS by a modified Delphi method. Scoring issues in CoMiSS included the tool’s widely variable predictability of CMA and the use of inappropriate stool consistency assessment and other symptoms.  The reasons for the panel’s decisions to keep the original scoring unchanged or to make changes to improve the sensitivity of CoMiSS were explained in the respective subsections of the manuscript. Based on the panel discussion, the authors concluded that CoMiSS should be updated by lowering the cut-off score from 12 to ≥10 to better predict the clinical diagnosis of CMA, extending the patient age up to 12 months, and including angioedema as an additional scoring symptom.

Diagnosing CMA is challenging since patients do not necessarily exhibit immediate severe reactions that are easily recognizable. Thus, it is important to raise the awareness of healthcare professionals as well as parents and caretakers about the signs and symptoms of potential CMA for early detection. The authors’ effort to evaluate and improve the current tool based on clinical evidence is therefore commended. The manuscript, however, seems to lack details on the methodology they used for their evaluation process and evidence supporting their decisions. Some specific issues and questions raised are listed below:

·         In the first paragraph of the introduction section (line 73), it would be helpful to the readers to provide the reasons why the diagnosis of CMA remains a challenge in clinical practice.

·         What were the differences in the studies that might be the cause of the wide range of sensitivity (20%-77%) of CoMiSS (line 104)? What were the infants' dietary habits (the percentages and frequencies of cow’s milk intake in relation to breast milk or other diets) in the studies on which CoMiSS is based?

·         The two short paragraphs (114-118) should be developed further to explain the potential issues relating to stool consistency and other symptoms.

·         “Atopic march” typically refers to the development of other atopic conditions such as asthma and rhinitis later in life, and not necessarily the development of additional allergies as the author referred (lines 126-127).

·         What was the “modified Delphi method”? What specific issues were raised, and what specific evidence supported the authors’ decisions for symptom inclusion and scoring?

·         In the “Crying/irritability” section, the authors state that objective measurement of the severity and duration of crying are unrealistic due to “cost and technical limitations.” However, it is not clear what these limitations are.

·         A “sick infant” in the hematochezia section is vague and should be further defined with distinguishing symptoms (line 253).

·         Angioedema was added to the updated CoMiSS. What would be the scoring for the newly added symptom?

·         The authors state that respiratory symptoms are acute and IgE-mediated during food challenges (lines 283-284). However, such symptoms must last over a week to be considered. It is unclear why persisting respiratory symptoms would be more indicative of CMA.

·         Does the recommendation include the quantification of total and/or allergen-specific IgE levels when a patient scores 10 with CoMiSS (line 391)?

Other minor issues include:

·         The abbreviation “CM,” should be defined independently from CMA (line 74).

·         The sentence, “Non IgE-mediated cow’s milk related symptoms might be non-IgE mediated CMA,” is redundant (also, consistent hyphenation should be used).

·         The legend for Table 1 describes the annotation marked with (§), but no such annotation is found in the table. Also, should “eye swelling and redness” be categorized under “respiratory” symptoms?

·         Why is GER bracketed as one of the GI symptoms in line 137?

·         An extra bracket after “growth” in line 139.

·         Other typos and grammatical errors are present in the manuscript.

Author Response

Reviewer 1

We thank the reviewer for the constructive comments and suggestions.

In the first paragraph of the introduction section (line 73), it would be helpful to the readers to provide the reasons why the diagnosis of CMA remains a challenge in clinical practice.

Additional information was added to the manuscript.

What were the differences in the studies that might be the cause of the wide range of sensitivity (20%-77%) of CoMiSS (line 104)?

The sensitivity of 20 % for the cut-off 12 was found in a study including infants with hematochezia /FPIAP performed in China.(1) Both conditions could affect the outcome, as the best cut-off of this cohort according to AUC was 6, and there are no data available on the CoMiSS performance in the healthy cohort. The best-documented sensitivity was found in a study with infants presenting with symptoms suggestive of CMA. Based on their clinical presentation, almost 89% (74 of 83) of the children were suspected of having CMPA.(2) Regarding this information, the perception of symptoms might play a role in patients' selection and influence the sensitivity level revealed in this study.

We added a summary of this information

1.             Vandenplas Y, Zhao ZY, Mukherjee R, Dupont C, Eigenmann P, Kuitunen M, et al. Assessment of the Cow's Milk-related Symptom Score (CoMiSS) as a diagnostic tool for cow's milk protein allergy: a prospective, multicentre study in China (MOSAIC study). BMJ Open. 2022 Feb 17;12(2):e056641.

2.             Prasad R, Venkata RSA, Ghokale P, Chakravarty P, Anwar F. Cow's Milk-related Symptom Score as a predictive tool for cow's milk allergy in Indian children aged 0-24 months. Asia Pac Allergy. 2018 Oct;8(4):e36

What were the infants' dietary habits (the percentages and frequencies of cow’s milk intake in relation to breast milk or other diets) in the studies on which CoMiSS is based?

Among the validation studies were two original studies without data on the type of feeding (the study by Prasad(2) and the Sirin - Kose study (3)). Furthermore, the pooled analysis study (4) did not identify the feeding type of included subjects. Two studies included infants fed with formula (both by Vandenplas ).(4,5) The studies on the tolerability of treatment formulas were performed in formula-fed cohorts. Among the studies assessing sensitivity and specificity were cohorts including exclusively breastfed infants and infants with combined feeding (breast and formula fed) having the percentage of infants fed with formula from 2.3% (the Selbuz study) (6) to 77% (the Salvatore study).(7)

The data on feeding frequencies were not taken into count.

We added a summary of this information

2.             Prasad R, Venkata RSA, Ghokale P, Chakravarty P, Anwar F. Cow's Milk-related Symptom Score as a predictive tool for cow's milk allergy in Indian children aged 0-24 months. Asia Pac Allergy. 2018 Oct;8(4):e36.

3.             Sirin Kose S, Atakul G, Asilsoy S, Uzuner N, Anal O, Karaman O. The efficiency of the symptom-based score in infants diagnosed with cow's milk protein and hen's egg allergy. Allergol Immunopathol (Madr). 2019 Jun;47(3):265–71.

4.             Vandenplas Y, Steenhout P, Järvi A, Garreau AS, Mukherjee R. Pooled Analysis of the Cow's Milk-related-Symptom-Score (CoMiSSTM) as a Predictor for Cow's Milk Related Symptoms. Pediatr Gastroenterol Hepatol Nutr. 2017 Mar;20(1):22–6.

5.             Vandenplas Y, Althera Study Group, Steenhout P, Grathwohl D. A pilot study on the application of a symptom-based score for the diagnosis of cow's milk protein allergy. SAGE Open Med. 2014;2:2050312114523423.

6.             Selbuz SK, AltuntaÅŸ C, Kansu A, KırsaçlıoÄŸlu CT, KuloÄŸlu Z, Ä°larslan NEÇ, et al. Assessment of cows milk-related symptom scoring awareness tool in young Turkish children. J Paediatr Child Health. 2020 May 29;

7.             Salvatore S, Bertoni E, Bogni F, Bonaita V, Armano C, Moretti A, et al. Testing the Cow's Milk-Related Symptom Score (CoMiSSTM) for the Response to a Cow's Milk-Free Diet in Infants: A Prospective Study. Nutrients [Internet]. 2019 Oct 8 [cited 2020 May 28];11(10). Available from: https://www.ncbi.nlm.nih.gov/pmc/articles/PMC6835327/

The two short paragraphs (114-118) should be developed further to explain the potential issues relating to stool consistency and other symptoms.

We adapted the structure of the paper, and brought this to the method section.

“Atopic march” typically refers to the development of other atopic conditions such as asthma and rhinitis later in life, and not necessarily the development of additional allergies as the author referred (lines 126-127).

This has been corrected.

What was the “modified Delphi method”? What specific issues were raised, and what specific evidence supported the authors’ decisions for symptom inclusion and scoring?

More information is included in the manuscript.

·         In the “Crying/irritability” section, the authors state that objective measurement of the severity and duration of crying are unrealistic due to “cost and technical limitations.” However, it is not clear what these limitations are.

We adapted the manuscript. In fact, since there is no device on the market, cost does not play a role. " However, cost, and technical limitations render such options unrealistic, as there is no de-vice on the market that accurately measures infant crying"

·         A “sick infant” in the hematochezia section is vague and should be further defined with distinguishing symptoms (line 253).

We added: a sick infant (painful abdomen, vomiting, pallor, lethargy, hypo- or hyperthermia, …)

Angioedema was added to the updated CoMiSS. What would be the scoring for the newly added symptom?

The addition of angio-edema does not change the scoring. This was added to the manuscript: If present, urticaria and angioedema both score 6; the combination of both also scores 6.

The authors state that respiratory symptoms are acute and IgE-mediated during food challenges (lines 283-284). However, such symptoms must last over a week to be considered. It is unclear why persisting respiratory symptoms would be more indicative of CMA.

We adapted the manuscript: Distinguishing between respiratory symptoms caused by infection or allergy remains challenging during infancy. Although most frequently respiratory symptoms during a challenge are of the acute type, it was estimated that acute respiratory symptoms during infancy are by far more frequently caused by infection than by an allergic reaction.

Does the recommendation include the quantification of total and/or allergen-specific IgE levels when a patient scores ≥10 with CoMiSS (line 391)?

The following has been added: Although a positive SPT or elevated sIgE levels increase the likelihood of CMA, these test are only indicative of sensitization and require an OFC to confirm the diagnosis [9,52].  

The abbreviation “CM,” should be defined independently from CMA (line 74).

Done

The sentence, “Non IgE-mediated cow’s milk related symptoms might be non-IgE mediated CMA,” is redundant (also, consistent hyphenation should be used).

We rephrased the sentence: Symptoms, clearly related to milk ingestion, may occur without any demonstrable immune mediated allergic mechanism or may  be due to non IgE mediated CMA.

The legend for Table 1 describes the annotation marked with (§), but no such annotation is found in the table. Also, should “eye swelling and redness” be categorized under “respiratory” symptoms?

We thank the reviewer to have noticed this. It was deleted.

Why is GER bracketed as one of the GI symptoms in line 137?

Because it is the abbreviation; but GER was not used later in the manuscript - so it was deleted

An extra bracket after “growth” in line 139.

Adapted

Other typos and grammatical errors are present in the manuscript.

We thank the reviewer for the effort done. We may have missed some, since we did not receive the version of the manuscript with the corrections made by the reviewer.

Reviewer 2 Report

The awareness tool CoMiSS® is interested by many researchers in the field of CMA. It is a good news that CoMiSS® is updating. However, I still have a few suggestions. My suggestions to strengthen the manuscript are included below:

(1) The structure of this manuscript could be improved. Please organize the section of Methods  and the section of Results section .

(2) Line 143-144, if irritability would be  classified as "other" organ system, then 52% of CMA-infants would present symptoms in other organ systems. That is not mentioned in reference 13. Reference 13 seems incorrect here.

(3) Line 177-178: In the modified Delphi, experts usually need to complete multiple rounds of surveys to achieve consensus. Therefore, please elaborate that how to use the modified Delphi to establish consensus on the statements?

Specific points are:

â‘  How many rounds of surveys were performed?

â‘¡ How to collect the comment from experts?

â‘¢ How to control the performing quality of the modified Delphi?

â‘£ How many statements should be voted?

⑤ How is eligible as a panel member?

â‘¥ How to communicate with the experts to eliminate the divergence comment?

(4) Table 2: Could excessive crying and irritability be classified as central nervous system symptoms? Because these symptoms are closely associated with CMA and could be classified more carefully. The symptoms of the central nervous system obviously are different from“other” organ systems.

(5) Table 3: The process of the group to consensus should be recorded.

For example:

â‘  Each round of voting results should be recorded

â‘¡ Each round of divergences of experts should be recorded.

Author Response

Reviewer 2.

We thank the reviewer for the suggestions. They have all been considered in the revised version of the manuscript.

The structure of this manuscript could be improved. Please organize the section of Methods  and the section of Results section .

We included a methods and results section

Line 143-144, if irritability would be  classified as "other" organ system, then 52% of CMA-infants would present symptoms in other organ systems. That is not mentioned in reference 13. Reference 13 seems incorrect here.

We thank the reviewer to have notice the error in the reference number. This has been corrected.

Table 2: Could excessive crying and irritability be classified as central nervous system symptoms? Because these symptoms are closely associated with CMA and could be classified more carefully. The symptoms of the central nervous system obviously are different from “other” organ systems.

We added that crying and irritability can be the consequence of gastrointestinal but also of skin manifestations

Suggestions regarding methodology.

Methodology is now included more extensively.

Round 2

Reviewer 1 Report

The authors have thoroughly addressed the issues raised.